# A Combined *Angelica gigas* and *Artemisia dracunculus* Extract Prevents Dexamethasone-Induced Muscle Atrophy in Mice through the Akt/mTOR/FoxO3a Signaling Pathway

**DOI:** 10.3390/cells11203245

**Published:** 2022-10-15

**Authors:** Hyun-Ji Oh, Heegu Jin, Byung-Yong Kim, Ok-Hwan Lee, Boo-Yong Lee

**Affiliations:** 1Department of Food Science and Biotechnology, College of Life Science, CHA University, Seongnam 13488, Korea; 2CH Labs, Seoul 07249, Korea; 3Department of Food Biotechnology and Environmental Science, Kangwon National University, Chuncheon 24341, Korea

**Keywords:** *Angelica gigas*, *Artemisia dracunculus*, dexamethasone-induced muscle atrophy, Akt/mTOR signaling pathway

## Abstract

Since skeletal muscle atrophy resulting from various causes accelerates the progression of several diseases, its prevention should help maintain health and quality of life. A range of natural materials have been investigated for their potential preventive effects against muscle atrophy. Here, ethanol extracts of *Angelica gigas* and *Artemisia dracunculus* were concentrated and dried, and mixed at a ratio of 7:3 to create the mixture CHDT. We then evaluated the potential for CHDT to prevent muscle atrophy and explored the mechanisms involved. CHDT was orally administered to C57BL/6 mice daily for 30 days, and dexamethasone (Dex) was intraperitoneally injected daily to induce muscle atrophy from 14 days after the start of oral administration. We found that CHDT prevented the Dex-induced reductions in muscle strength, mass, and fiber size, likely by upregulating the Akt/mTOR signaling pathway. In addition, CHDT reduced the Dex-induced increase in the serum concentrations of pro-inflammatory cytokines, which directly induce the degradation of muscle proteins. These findings suggest that CHDT could serve as a natural food supplement for the prevention of muscle atrophy.

## 1. Introduction

Skeletal muscle is an abundant tissue in the human body, accounting for more than 40% of the total body mass, and plays important roles not only in physical exercise, but also in metabolism, such as that of glucose and lipids [1,2,3]. Skeletal muscle atrophy, caused by factors such as malnutrition, injury, and aging, reduces quality of life and can lead to the development of various metabolic diseases. It can also progress to sarcopenia, defined by reductions in both muscle mass and strength. Sarcopenia is one of the major factors responsible for the shortening of the healthy lifespan of older people and greatly reduces quality of life by causing a loss of mobility [4]. The prevention of muscle atrophy and the maintenance of healthy muscle are essential to provide the energy necessary for physical activity and to prevent metabolic disorders, permitting healthy aging.

Muscle mass is maintained through a balance between muscle protein synthesis and breakdown, and muscle atrophy begins when this balance is disrupted, involving a decrease in intramuscular protein metabolism and/or an acceleration of muscle protein degradation [5,6]. The Akt/mechanistic target of rapamycin (mTOR) signaling pathway is known to be closely related to muscle hypertrophy and atrophy: Akt promotes protein synthesis by activating mTOR and its downstream substrates in muscle cells [7,8]. The mTOR complex is biochemically distinguished into mTOR complex 1 (mTORC1) and mTOR complex 2 (mTORC2). Both complexes have mTOR as a common subunit, each with its own unique components. Among them, mTORC1 is known to be a key regulator of skeletal muscle mass. mTORC1 upregulates protein translation through the phosphorylation of eukaryotic translation initiation factor 4E-binding protein 1 (*4EBP1*) and ribosomal protein S6 kinase beta-1 (*S6K1*), leading to cell growth and proliferation [9,10]. In addition, myoblast determination protein 1 (*MyoD*) and myogenin, which are myogenic transcription factors, reduce muscle atrophy [11,12].

A major mediator of muscle atrophy is forkhead box O3 (FoxO3a), which increases proteolysis through activation of the muscle-specific E3 ubiquitin ligases F-box protein (*Fbx32*, also known as atrogin) and muscle ring-finger 1 (MuRF1) [13]. FoxO-mediated proteolysis is inhibited by Akt because it phosphorylates FoxO3a, thereby preventing its nuclear translocation and activity as a transcription factor [14]. Thus, the pharmacologic activation of Akt/mTOR signaling, a key promoter of protein synthesis, and the inhibition of FoxO3a by Akt, preventing muscular atrophy, represent attractive approaches to the prevention of muscle wasting. In addition, pro-inflammatory cytokines that promote the degradation of myofibrillar proteins and reduce protein synthesis, thereby exacerbating muscle wasting, are other major causes of muscle atrophy [15]. Pro-inflammatory cytokines such as interleukin (IL)-1, IL-6, and tumor necrosis factor (TNF)-α induce ubiquitin-dependent protein degradation and cell death, and therefore interventions targeting the production of pro-inflammatory cytokines may also be effective means of ameliorating muscle atrophy [16].

Glucocorticoids (GCs) are steroid hormones that are widely used to treat conditions such as autoimmune diseases and cancer. However, high doses and the long-term use of GCs are associated with serious side effects, such as hyperglycemia, osteoporosis, and muscle atrophy [17,18]. Persistently high concentrations of GCs are associated with catabolic effects in skeletal muscle and pathological states characterized by muscle atrophy, including sepsis, cachexia, and diabetes [18,19]. Because of these effects, the long-term administration of a high dose of dexamethasone (Dex), a synthetic GC, is used to induce muscle atrophy in animal models. Previous studies have shown that Dex both suppresses protein synthesis and induces protein degradation in the muscles of mice [20].

Angelica gigas is a well-known medicinal herb that is used in Asian countries. It has been used for the treatment of anemia, for its anti-inflammatory effects, and for the healthcare of women [21,22,23]. In particular, decursin, a principal component of *A. gigas*, has beneficial effects on circulatory disease and antioxidant effects [24,25]. *Artemisia dracunculus* has also been demonstrated in several studies to be bioactive, having anti-inflammatory, hepatoprotective, and antihyperglycemic effects [26,27,28]. Although several beneficial effects of *A. gigas* and *A. dracunculus* have been demonstrated, their potential for the protection of muscle atrophy has not been studied in detail. In the present study, we prepared a mixture of extracts of A. gigas and A. dracunculus (named CHDT) and evaluated its efficacy for the prevention of muscle atrophy in mice with Dex-induced muscle atrophy and explored the mechanism involved.

## 2. Materials and Methods

### 2.1. Preparation of CHDT

The A. gigas and A. dracunculus used in the present study were purchased from Hangaram GF Co., Ltd. (Seoul, Korea). A. gigas and A. dracunculus were extracted using 70% (*v*/*v*) ethanol at 75 °C for 12 h, and the extracts obtained were concentrated and dried. The powdered extracts of A. gigas and A. dracunculus were then mixed at a ratio of 7:3, and this mixture is referred to as CHDT (CH Labs Danggui Tarragon). The dietary composition of the CHDT used in the study is shown in Table 1. CHDT used in this study was obtained from CH Labs (Seoul, Korea). The CHDT powder was dissolved in DW for oral administration to mice.

High-performance liquid chromatography (HPLC) was performed using a Waters E2695 Separation Module HPLC (Milford, MA, USA) and a PDA (Photodiode Array) detector (Waters). The CHDT was separated on a YMC Triart C18 (250 × 4.6 mm, 5 μm) (YMC, Sungnam, Korea) after dissolution in methanol and injection in a volume of 10 μL. The standards (decursin and 7-methoxycoumarin) for the HPLC analysis were obtained from Sigma Aldrich (St. Louis, MO, USA). For the measurement of decursin, the mobile phases were distilled water (DW) containing 0.1% (*v*/*v*) trifluoroacetic acid (TFA) (solvent A) and acetonitrile (ACN) containing 0.1% (*v*/*v*) TFA (solvent B). The gradient program comprised 0–50 min, 5–80% B; 50–55 min, 80% B; 55–56 min, 80–5% B; and 56–70 min, 5% B; a flow rate of 0.7 mL/min was used; and UV detection was performed at 330 nm. To measure 7-methoxycoumarin, the mobile phase was composed of DW containing 0.1% TFA (solvent A) and ACN (solvent B). The gradient program was composed of 0–5 min, 20% B; 5–20 min, 20–100% B; 20–30 min, 100% B; 30–31 min, 100–80% B; 31–40 min, 80% B; 40–50 min, 80–20% B; and 50–60 min, 20% B; a flow rate of 1.0 mL/min was used; and UV detection was performed at 322 nm.

### 2.2. Animals and Treatments

Six-week-old C57BL/6J mice were purchased from the Korea Research Institute of Bioscience and Biotechnology (Daejeon, Korea). The animal experiments were approved by the Institutional Animal Care and Use Committee of CHA University (IACUC210135). The mice were housed under a 12 h light/dark cycle and allowed 1 week to acclimatize to their new environment. They were randomly assigned to four groups (n = 10/group) as follows: (i) a control group (CON), (ii) mice intraperitoneally (IP) injected with 1 mg/kg/day Dex (DEX), iii) mice administered 350 mg/kg/day CHDT and IP injected with 1 mg/kg/day Dex (DEX+CHDT350), and iv) mice administered 500 mg/kg/day CHDT and IP injected with 1 mg/kg/day Dex (DEX+CHDT500). After the week of adaptation, CHDT (350 or 500 mg/kg/day) was orally administered to the appropriate mice by gavage daily for 30 days, commencing 15 days before Dex injection commenced, for a period of 15 days. At the end of the study, the mice were sacrificed, and blood and tissue samples were collected and stored at −80 °C until further analysis.

### 2.3. Reagents

Water-soluble Dex was purchased from Sigma Aldrich (St. Louis, MO, USA). Antibodies specific for phosphoinositide 3-kinase (PI3K), phospho-4EBP1, 4EBP1, phospho-ribosomal protein S6 kinase beta-1 (p70S6K1), p70S6K1, and myogenin were purchased from Santa Cruz Biotechnology (Dallas, TX, USA). Antibodies specific for phospho-FoxO3a, FoxO3a, phospho-Akt, Akt, phospho-mammalian target of rapamycin (mTOR), and mTOR were purchased from Cell Signaling Technology (Danvers, MA, USA). Antibodies specific for MyoD, Fbx32, and MuRF1 were purchased from Abcam (Cambridge, UK).

### 2.4. Measurement of Body Mass and Grip Strength

The body mass and muscle (grip) strength of mice in all the groups were measured on days 0, 7, 14, 18, 22, 26, and 30 of the study period. Grip strength was measured using a Chatillon force measurement system (Columbus Instruments, Columbus, OH, USA) by placing each mouse on a mesh grid and allowing it to grasp the net with all four feet, after which its tail was pulled three-to-five times and the maximum force was measured just before the mouse fell off the net. The mean force measured for each mouse was recorded.

### 2.5. Measurement of Lean Mass

The lean mass of the mice was measured twice: before the start of the study and at the end of the period of administration (on day 30). The lean mass of the mice was assessed using dual energy X-ray absorptiometry (DXA) with an InAlyzer dual X-ray absorptiometer (Medikors, Gyeonggi, Korea). The body composition of each mouse was assessed using the device software (lean, fat, and bone tissue). The mouse hindlimb was selected as the region of interest (ROI). The percentage lean mass of each mouse was calculated by normalizing the hindlimb lean mass to the total body mass.

### 2.6. Western Blot Analysis

Samples of the quadriceps (QUA) and gastrocnemius (GAS) muscles were lysed in lysis buffer (iNtRON Biotechnology, Seoul, Korea) containing protease and phosphatase inhibitors. The total protein contents of the lysates were quantified using a BCA protein assay (Pierce, Rockford, IL, USA). Lysates containing equal amounts of protein were separated by SDS-PAGE and the separated proteins were transferred to membranes. The membranes were blocked with 5% skimmed milk and washed with Tris-buffered saline containing 0.05% Tween-20. The membranes were incubated overnight at 4 °C with the primary antibodies; PI3K, p-Akt, Akt, p-mTOR, mTOR, p-p70S6K, p70S6K, p-4EBP1, 4EBP1, myoD, myogenin, p-FoxO3a, FoxO3a, Fbx32, and MuRF1, and then with a secondary antibody (peroxidase-conjugated anti-rabbit, anti-mouse, or anti-goat antibodies; Bio-Rad, Hercules, CA, USA) for 1 h at room temperature. The target protein bands were detected using a EZ-Western Lumi Femto kit (DoGenBio, Seoul, Korea) and imaged using a LAS-4000 instrument (GE Healthcare Life Sciences, Marlborough, MA, USA). The relative band intensities were quantified using ImageJ software (NIH, Bethesda, MD, USA).

### 2.7. Histological Analysis

QUA and GAS muscles removed from the hindlimbs of the mice were fixed in 4% paraformaldehyde. The samples were then embedded in paraffin blocks, sectioned at 10 μm, and hematoxylin and eosin (H&E)-stained. The muscle sections were then examined at ×200 magnification using a Nikon E600 microscope (Nikon, Tokyo, Japan). The cross-sectional areas (CSAs) of the muscle fibers were measured using ImageJ software.

### 2.8. Serum Analysis

Blood samples were obtained by cardiac puncture at the time of sacrifice and centrifuged at 800× *g* for 15 min at 4 °C to obtain serum. The serum concentrations of IL1, IL-6, and TNF-α were measured using a Milliplex MAP Mouse High-sensitivity T-cell Panel (Merck Millipore, Burlington, MA, USA). The serum activities of aspartate transaminase (AST) and alanine transaminase (ALT) were analyzed using a Mouse AST ELISA (ab263882; Abcam, Cambridge, UK) and Alanine Transaminase Activity Assay (ab105134; Abcam, Cambridge, UK) kits, respectively, according to the manufacturer’s protocol. A Luminex 100 analyzer (Luminex, Austin, TX, USA) was used for absorbance measurements. All the measurements were made in duplicate.

### 2.9. Statistical Analysis

All data are expressed as the mean ± standard error of the mean (SEM), and comparisons between groups were made using one-way analysis of variance (ANOVA) followed by Tukey’s test. *p* < 0.05 was accepted as indicating statistical significance.

## 3. Results

### 3.1. Analysis of CHDT

A representative HPLC chromatogram for the CHDT is shown as Figure 1. The peaks of decursin (the index component of A. gigas) and 7-methoxycoumarin (the index component of A. dracunculus) were identified, and their concentrations were analyzed. The concentrations of decursin and 7-methoxycoumarin in the CHDT were found to be 30.32 mg/g and 0.33 mg/g, respectively. The results of the analysis of the phenolic compounds and flavonoids in the CHDT are shown in Appendix A.

### 3.2. CHDT Ameliorates the Dex-Induced Loss of Body Mass and Muscle Strength of the Mice

To evaluate the effect of CHDT on Dex-induced muscle atrophy, CHDT was orally administered daily to 6-week-old mice for 30 days, and from 14 days after the start of administration, Dex was IP injected daily. The body mass and grip strength of the mice were measured on days 0, 7, 14, 18, 22, 26, and 30 of the 30 day study period, after which the mice were sacrificed for further tissue analysis (Figure 2A). The body mass did not significantly differ among the groups prior to the first Dex injection. Dex injection caused all three of the treated groups to lose weight. The DEX group showed gradual weight loss compared to the CON group, but this weight loss was significantly reduced by CHDT administration (Figure 2B). Similarly, there were no significant differences in the grip strengths of the groups prior to Dex injection. After the injection commenced, the grip strength of the DEX group was significantly lower than that of the CON group, but those of the CHDT-administered groups were maintained (in the DEX+CHDT350 group) or increased (in the DEX+CHDT500 group) (Figure 2C). These results indicate that the administration of CHDT may help prevent and/or reduce the losses of body mass and muscle strength associated with muscle atrophy.

### 3.3. CHDT Inhibits the Dex-Induced Loss of Lean Mass and Muscle Mass

To determine the effect of CHDT on muscle mass, the lean mass of the mice was measured using DXA before the start and at the end of the administration of CHDT (Figure 3A). The lean mass of the CON group in the ROI was slightly higher at the end point than at the start of the study, but there was no significant difference in the percentage lean mass in the ROI. This implies that the mice grew normally during the study period. The lean mass and percentage in the ROI of the DEX group were lower than those of the CON group, but the CHDT-administered groups showed dose-dependently higher values (Figure 3B,C).

We also measured the sizes and masses of the QUA and GAS muscles after their excision from the hindlimbs of the mice. The masses of both muscles were normalized to the most recently measured values of body mass. Consistent with the results of the DXA analysis, the sizes and masses of both muscles were significantly lower in the DEX group than in the CON group. However, the sizes and masses of both muscles in the CHDT-administered groups were greater than or similar to those of the CON group (Figure 3D,E). Taken together, these data suggest that the administration of CHDT prevents the Dex-induced reduction in the muscle mass of mice.

### 3.4. CHDT Activates the Akt/mTOR Signaling Pathway

Muscle protein synthesis is increased through activation of the Akt/mTOR signaling pathway [7]. To investigate the effect of CHDT on protein synthesis, which is reduced by Dex, we evaluated the expression and activation of components of the Akt/mTOR signaling pathway in the QUA and GAS muscles of the mice. We found that the protein expression of PI3K and the p-Akt/Akt and p-mTOR/mTOR ratios of both muscles in the DEX group were significantly lower than those of the CON group, but these ratios were increased by CHDT administration. The lower level of activation of mTOR induced by Dex administration was associated with lower levels of phosphorylation of the downstream mediators 4EBP1 and p70S6K. However, the p-4EBP1/4EBP1 and p-p70S6K/p70S6K ratios were significantly higher in the DEX+CHDT350 and DEX+CHDT500 groups (Figure 4A,B). These results indicate that CHDT can prevent the Dex-induced reduction in muscle protein synthesis in mice via the Akt/mTOR signaling pathway.

### 3.5. CHDT Restores the Expression of Myogenic Transcription Factors and Muscle Fiber Size in Dex-Treated Mice

Myogenesis occurs in normal young mice, which requires muscle synthesis [29]. To determine whether CHDT upregulates myogenesis, we measured the expression of myogenic transcription factors in the QUA and GAS muscles of each group of mice. The expression of MyoD and myogenin in both muscles of the DEX group was lower than that of the CON group. However, the expression of all these proteins in the 500 mg/mg/day CHDT-administered group was as high as that in the CON group (Figure 5A,B). This suggests that CHDT inhibits the Dex-related reduction in the expression of myogenic transcription factors. In addition, we determined the effect of CHDT on muscle fiber size. Consistent with the findings described above, histological analysis of the QUA and GAS muscles showed that the mean fiber size of each was smaller in the DEX group than in the CON group, and high-dose CHDT prevented the reduction in muscle fiber size caused by Dex injection (Figure 5C,D). Taken together, these findings imply that the administration of CHDT prevents the inhibition of myogenesis and the reduction in muscle fiber size induced by Dex that contribute to muscle atrophy.

### 3.6. CHDT Ncreases the Hosphorylation of FoxO3a and Reduces the Circulating Concentrations of Pro-Inflammatory Cytokines in Dex-Treated Mice

One of the most effective ways of preventing muscle atrophy is to inhibit the activation of mediators of muscle breakdown. Therefore, we measured the expression of FoxO3a, which is a key mediator of muscle wasting. When phosphorylated by Akt, FoxO3a is unable to undergo nuclear translocation from the cytoplasm, and therefore cannot upregulate the expression of genes encoding components of the ubiquitin-proteasome system. The ratio of p-FoxO3a/FoxO3a was lower in both muscles of the DEX group than in those of the CON group, but higher in the high-dose CHDT-administered group. This implies that the restoration of Akt expression by CHDT promotes the phosphorylation of FoxO3a, such that CHDT affects the nuclear translocation of FoxO3a. We also measured the expression of E3 ubiquitin ligases (Fbx32 and MuRF1), which play important roles in skeletal muscle atrophy. The expression of Fbx32 and MuRF1 was higher in both muscles of mice in the DEX group than in those of mice in the CON group, but the expression of both was lower in the CHDT-administered groups (Figure 6A,B). This suggests that CHDT may inhibit muscle wasting by reducing transcription by FoxO3a, thereby reducing protein ubiquitination.

Another major cause of muscle atrophy is greater production of pro-inflammatory cytokines, which directly cause muscle wasting [30]. Dex treatment significantly increased the serum concentrations of IL-1, IL-6, and TNF-α, but the administration of CHDT reduced the concentrations of all three cytokines at both doses (Figure 6C). Thus, CHDT may also ameliorate muscle atrophy by preventing the excessive production of pro-inflammatory cytokines.

## 4. Discussion

Skeletal muscle plays an important role in movement, energy metabolism, and normal healthy living. Muscle atrophy is a feature of various diseases and causes abnormal movement or restriction of activity, leading to poor exercise performance, and predisposes toward serious metabolic diseases [3,31]. Therefore, the development of means of preventing muscle atrophy should improve quality of life and survival. Drugs known to ameliorate muscle atrophy, such as testosterone and ghrelin, are limited in their clinical applications because of concerns regarding potential side effects [32]. In the search for safe and effective methods of preventing muscle atrophy, natural substances represent cost-effective sources of potential health benefits.

In the present study, A. gigas and A. dracunculus derivatives, which are widely used as natural treatments and are known to have several pharmacological activities, were studied. Analysis of the phenolic constituents of CHDT (Appendix A) and the evaluation of hepatotoxicity (Appendix A) suggests that this mixture may have beneficial effects and be safely administered. Because high-dose, continuous Dex treatment induces skeletal muscle atrophy by upregulating protein catabolism, we studied a mouse model of Dex-induced muscle atrophy. To evaluate the preventive effect of CHDT on Dex-induced muscle atrophy, CHDT was orally administered daily from 14 days before the injection of Dex commenced. The results suggest that the pre-administration of CHDT prevents weight loss and the loss of muscle strength in Dex-treated mice. In addition, body composition analysis and muscle mass measurement provided clear evidence that the substantial muscle loss induced by Dex was prevented by CHDT administration, especially at the higher dose. Based on these results, we next attempted to establish the mechanisms whereby CHDT might affect muscle protein synthesis and degradation.

Several previous studies have shown that the sustained administration of a high dose of Dex inhibits protein synthesis signaling and upregulates proteolytic signals in muscle [33,34,35,36]. The activation of Akt signaling, a major pathway for protein synthesis, has positive effects on muscle growth [37]. In the present study, we found that the phosphorylation of Akt was lower in the DEX group than in the CON group, implying that a reduction in Akt signaling is a feature of Dex-induced muscle atrophy. This was associated with lower phosphorylation of the Akt target mTOR and downstream intermediates that promote protein synthesis. However, CHDT administration dose-dependently increased Akt phosphorylation and prevented the impairments in downstream signaling of mTOR. Activation of S6K1 and 4EBP1 by CHDT administration resulted from regulation of mTORC1, implying that CHDT contributed to upregulating protein translation and muscle cell proliferation in skeletal muscle. These findings suggest that CHDT may prevent atrophy by stimulating muscle protein synthesis secondary to activation of the Akt/mTOR signaling pathway in mice. This effect may also be mediated through the prevention of the reduction in activity of the myogenic transcription factors MyoD and myogenin, causing a restoration of muscle fiber size. We found that Dex treatment suppressed the expression of MyoD and myogenin, but was significantly increased by CHDT treatment. These results were seen in both muscles, especially in GAS rather than QUA.

The inhibition of Akt activation is also associated with the induction of muscle atrophy through dephosphorylation of FoxO3a and consequent promotion of its nuclear translocation [38]. FoxO3a-induced protein degradation is mediated through the ubiquitin-proteasome and autophagy-lysosomal system. [39,40]. Adequate autophagy is known to be a required process for maintaining muscle mass [41], but hyperactivity of catabolic process, including the ubiquitin-proteasome system and autophagy-lysosomal system, leads to muscle wasting [42]. High expression of the E3 ubiquitin ligases Fbx32 and MuRF1 is a major mediator of muscle wasting and is highly correlated with the promotion of autophagy in muscle cells [13,43,44]. As expected, expression of these two proteins was increased by Dex treatment, but CHDT increased the phosphorylation of FoxO3a, which would block its nuclear translocation, and reduced the expression of E3 ubiquitin ligases, thereby downregulating both systems involved in muscle degradation. Taking all these findings together, it seems that CHDT may prevent muscular atrophy through effects on the Akt/mTOR and Akt/FoxO3a pathways, which participate in both protein synthesis and degradation in skeletal muscle.

Inflammation is another major cause of muscle atrophy, and pro-inflammatory cytokines are well known to induce muscle wasting by upregulating ubiquitin ligases and inducing proteolysis [45]. We found that Dex treatment increased the serum concentrations of the pro-inflammatory cytokines IL-1, IL-6, and TNF-a, but that these were significantly reduced by CHDT treatment. These results indicate that CHDT may also prevent Dex-induced muscle atrophy by reducing the production of pro-inflammatory cytokines.

In conclusion, Dex treatment likely induces muscle atrophy, characterized by losses of muscle mass and strength, through the Akt/mTOR and Akt/FoxO3a signaling pathways in mice, but the administration of CHDT prevents its deleterious effects on almost all the measured parameters, especially at a dose of 500 mg/kg/day. We have provided evidence that CHDT may both promote the synthesis of muscle protein and inhibit its degradation. Therefore, we propose that CHDT may represent a safe and effective means of preventing muscle atrophy. Clinical studies are now needed to evaluate the efficacy of CHDT in patients with muscle dysfunction or atrophy.

## Figures and Tables

**Figure 1 cells-11-03245-f001:**
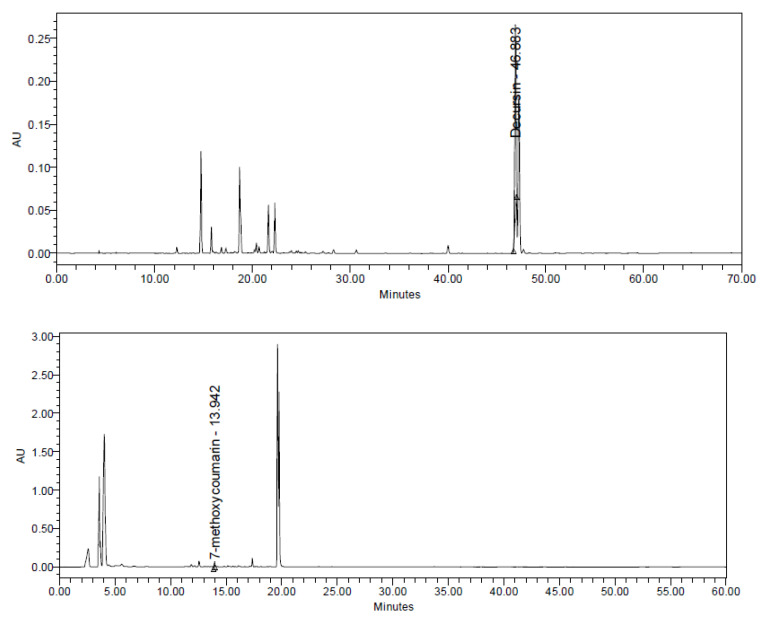
High-performance liquid chromatographic traces for the analysis of the A. gigas (**top**) and A. dracunculus (**bottom**) extracts.

**Figure 2 cells-11-03245-f002:**
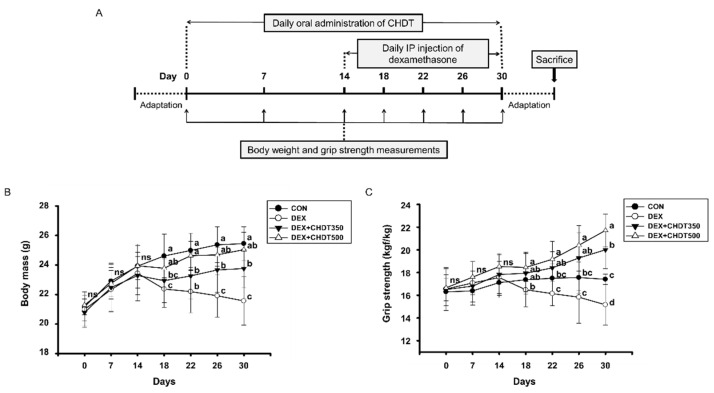
Effects of CHDT on the body mass and muscle strength of mice with Dex-induced muscle atrophy. (**A**) Experimental protocol. CHDT was orally administered daily to 6-week-old mice for 30 days, and from 14 days after the start of administration, Dex was IP injected daily. (**B**) Body mass and (**C**) grip strength of the mice were measured on days 0, 7, 14, 18, 22, 26, and 30. “ns” indicates that there was no statistically significant difference between all groups. (*p* > 0.05). Different letters indicate statistically significant differences; *p* < 0.05, a > b > c > d.

**Figure 3 cells-11-03245-f003:**
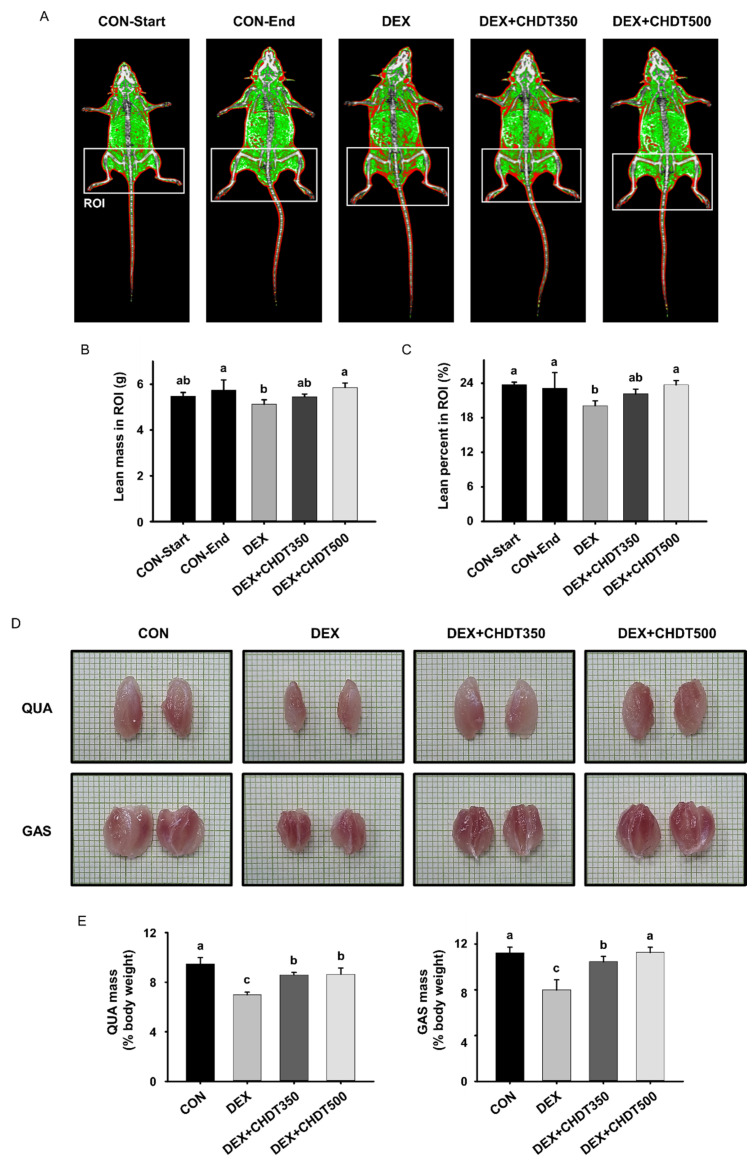
Effects of CHDT on the lean mass and muscle mass of mice with Dex-induced muscle atrophy. (**A**) Body composition images obtained using DXA analysis. Lean tissue is shown in green, fat in red, and bone in white. The lean mass of the mice was measured twice: before the start of the study and at the end of the period of administration (on day 30). (**B**) Lean mass and (**C**) lean percentage in the ROI, which surrounds the mouse hindlimbs. (**D**) At the end of the study, representative images of the quadriceps (QUA, top) and gastrocnemius (GAS, bottom) muscles were taken. The scale of one grid is 1 mm. (**E**) Masses of the QUA (left) and GAS (right) muscles, normalized to the most recently measured body mass. Different letters indicate statistically significant differences; *p* < 0.05, a > b > c.

**Figure 4 cells-11-03245-f004:**
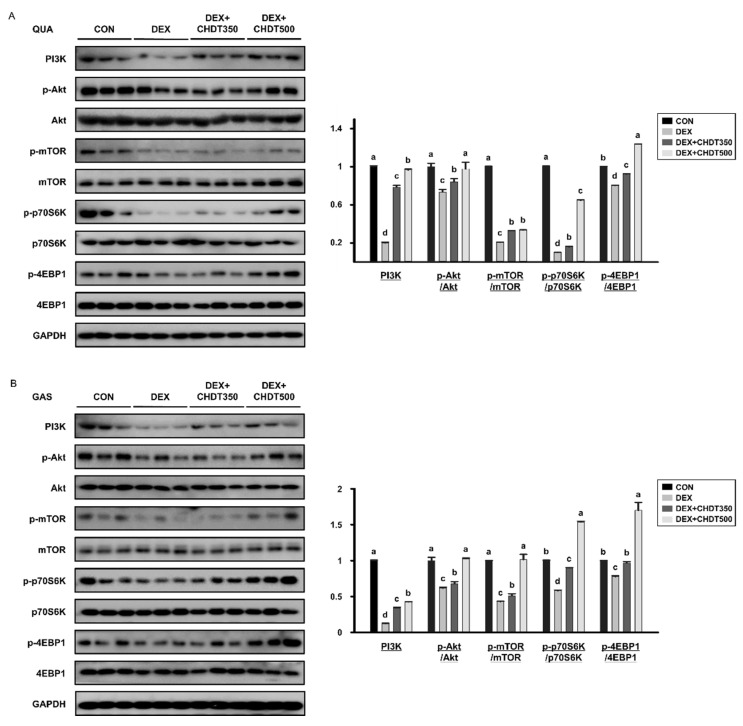
Effects of CHDT on the Akt/mTOR signaling pathway in mice with Dex-induced muscle atrophy. The protein expression levels of components of the Akt/mTOR signaling pathway were measured by Western blotting in (**A**) QUA and (**B**) GAS muscles. GAPDH was used as the loading control and the phosphorylation of the intermediates was normalized to the total expression levels of each protein. Different letters indicate statistically significant differences; *p* < 0.05, a > b > c > d.

**Figure 5 cells-11-03245-f005:**
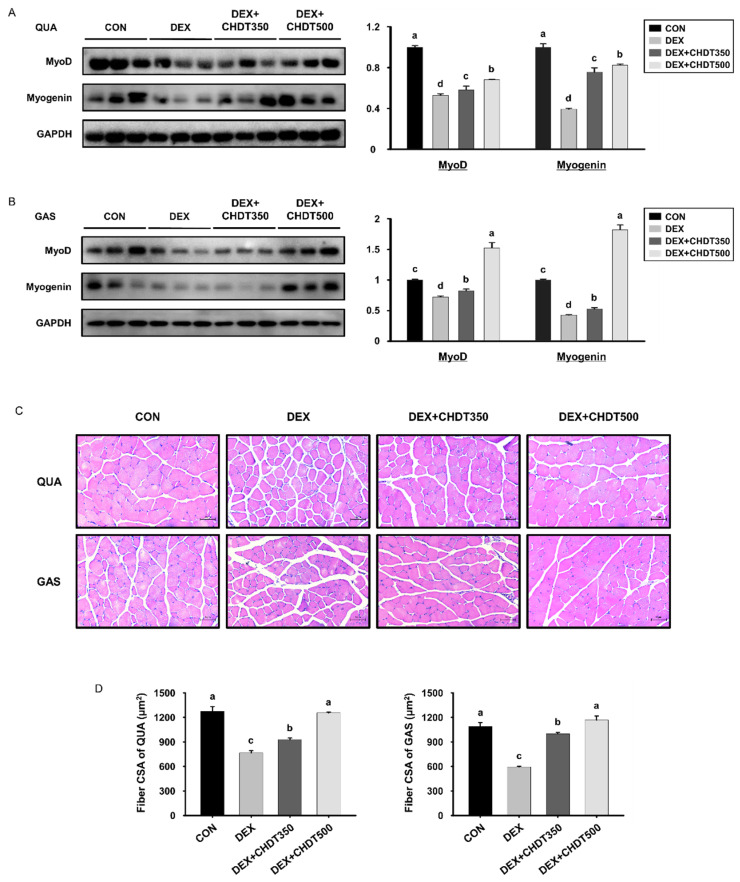
Effects of CHDT on the expression of myogenic transcription factors and muscle fiber size in mice with Dex-induced muscle atrophy. The protein expression levels of myogenic transcription factors were measured by Western blotting in (**A**) QUA and (**B**) GAS muscles. (**C**) Hematoxylin and eosin-stained transverse sections of the QUA (top) and GAS (bottom) muscles. (**D**) Cross sectional area (CSA) of QUA (left) and GAS (right) muscles.Different letters indicate statistically significant differences; *p* < 0.05, a > b > c > d.

**Figure 6 cells-11-03245-f006:**
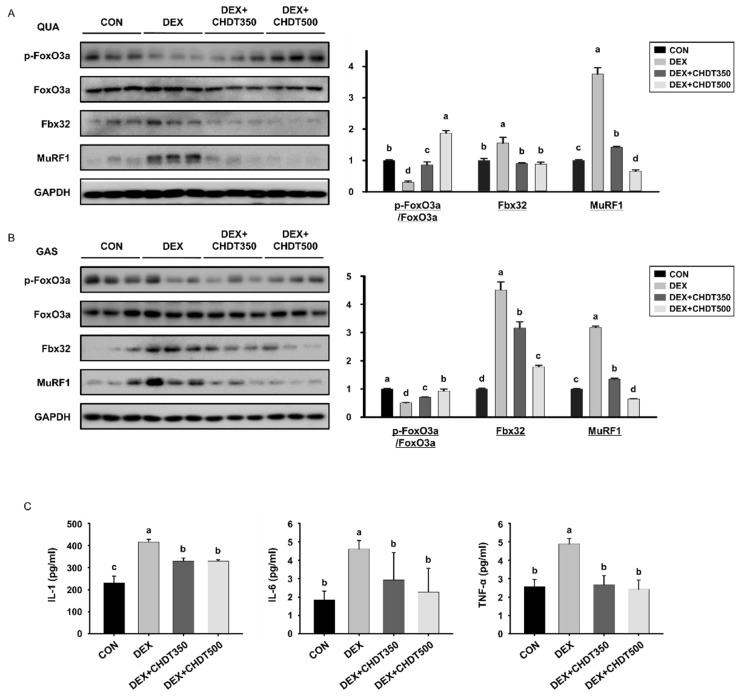
Effects of CHDT on the protein expression of FoxO3a and E3 ubiquitin ligases and the serum concentrations of pro-inflammatory cytokines. The protein expression levels of FoxO3a and E3 ubiquitin ligases were measured by Western blotting in (**A**) QUA and (**B**) GAS muscles. GAPDH was used as the loading control and the phosphorylation of FoxO3a was normalized to the total expression of the protein. (**C**) Serum concentrations of pro-inflammatory cytokines. Different letters indicate statistically significant differences; *p* < 0.05, a > b > c > d.

**Table 1 cells-11-03245-t001:** Composition of the CHDT.

Component	Content (%)
Crude protein	3.22
Carbohydrate	83.70
Crude fat	5.80
Ash	4.17
Moisture	3.11
Total	100

## Data Availability

Not applicable.

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
