# Peer review of "A Combined Angelica gigas and Artemisia dracunculus Extract Prevents Dexamethasone-Induced Muscle Atrophy in Mice through the Akt/mTOR/FoxO3a Signaling Pathway"

_cells, 2022, doi:10.3390/cells11203245_

Round 1

Reviewer 1 Report

Oh et al. reported the beneficial effects of the combination of Angelica gigas and Artemisia dracunculus extract on dexamethasone-induced muscle atrophy.

1. Why was pre-administration of CHDT for 15 days required? The authors need to show whether CHDT still prevents DEX-induced muscle atrophy without pre-administration of CHDT.

2. Several papers show that Dex enhances muscle regeneration and decreases the expression of inflammatory factors, which is contradictory to your observation (Cell Death Discov. 7, 35 (2021); Neuromuscul Disord. 2010 Feb;20(2):111-21). The authors need to test the effect of CHDT on C2C12 myogenesis in the presence of Dex.

3. mTORC1 regulates protein synthesis by activating S6K1 and 4EBP1. Since CHDT activated Akt, not S6K1 and 4EBP1, p-mTOR might come from mTORC2. The authors need to test whether CHDT increases mTORC2 activity in DEX-induced muscle atrophy by immunoprecipitating with an anti-rictor and testing mTOR phosphorylation in IPed mTOR. 

4. The authors need to test the effect of rapamycin on CHDT-induced prevention of DEX-induced muscle.  

4. When did you measure the lean mass and muscle mass of mice with DEX-induced muscle atrophy in Fig. 3? You need to add detailed information (duration for treatment; the number of mice for a group) in all Figure legends.

5. The effect of CHDT on MyoD and myogenin in Qua was different from one in Gas. The authors need to discuss this in ‘Discussion’ section. 

6. The authors need to move the result of HPLC from the materials and methods section to the result section. 

7. All data points need to be presented in all figures.

Author Response

We sincerely thank the Reviewer for their thorough review of our manuscript and for the constructive suggestions that we received. We have tried to adequately respond to each comment received from the Reviewer. We believe that the Reviewer’s comments and suggestions have considerably improved our manuscript. We hope that our manuscript will be published in the Cells.

Comments from Reviewer 1

Point 1: Why was pre-administration of CHDT for 15 days required? The authors need to show whether CHDT still prevents DEX-induced muscle atrophy without pre-administration of CHDT.

Response 1: Thanks for your comment. The reason we pro-administered CHDT is to investigate the “protective and preventive effects of CHDT” on muscle atrophy accompanying malnutrition, injury, aging, and disease. This study demonstrated that the pre-administration of CHDT before the onset of the muscle atrophy effectively prevented the development of severe muscle atrophy. In addition, in Figure 2C, the CHDT administration group showed a continuous increase in muscle strength despite the dexamethasone treatment.

Point 2: Several papers show that Dex enhances muscle regeneration and decreases the expression of inflammatory factors, which is contradictory to your observation (Cell Death Discov. 7, 35 (2021); Neuromuscul Disord. 2010 Feb;20(2):111-21). The authors need to test the effect of CHDT on C2C12 myogenesis in the presence of Dex.

Response 2: Thanks for your opinion. The papers you referenced used C2C12 cells treated with low dose of Dex, but our study used a mouse model treated with continuous and high dose of Dex. There are many studies that continuous and high-dose Dex treatment induces muscle atrophy and increases inflammation [1-5]. In fact, the mouse model of muscle atrophy by Dex treatment is widely used for muscle-related research. In all parameters of our data, the DEX group showed a phenotype of muscular atrophy compared to the CON group, which means that a mouse model of muscular atrophy caused by Dex treatment was established.

  1. Wang, L.; Jiao, X. F.; Wu, C.; Li, X. Q.; Sun, H. X.; Shen, X. Y.; Zhang, K. Z.; Zhao, C.; Liu, L.; Wang, M.; Bu, Y. L.; Li, J. W.; Xu, F.; Chang, C. L.; Lu, X.; Gao, W., Trimetazidine attenuates dexamethasone-induced muscle atrophy via inhibiting NLRP3/GSDMD pathway-mediated pyroptosis. Cell Death Discov 2021, 7, (1), 251.
  2. Otsuka, Y.; Egawa, K.; Kanzaki, N.; Izumo, T.; Rogi, T.; Shibata, H., Quercetin glycosides prevent dexamethasone-induced muscle atrophy in mice. Biochem Biophys Rep 2019, 18, 100618.
  3. Huang, Y.; Chen, K.; Ren, Q.; Yi, L.; Zhu, J.; Zhang, Q.; Mi, M., Dihydromyricetin Attenuates Dexamethasone-Induced Muscle Atrophy by Improving Mitochondrial Function via the PGC-1alpha Pathway. Cell Physiol Biochem 2018, 49, (2), 758-779.
  4. Ma, K.; Mallidis, C.; Bhasin, S.; Mahabadi, V.; Artaza, J.; Gonzalez-Cadavid, N.; Arias, J.; Salehian, B., Glucocorticoid-induced skeletal muscle atrophy is associated with upregulation of myostatin gene expression. Am J Physiol Endocrinol Metab 2003, 285, (2), E363-71.
  5. Mills, C. M.; Marks, R., Side effects of topical glucocorticoids. Curr Probl Dermatol 1993, 21, 122-31.

Point 3: mTORC1 regulates protein synthesis by activating S6K1 and 4EBP1. Since CHDT activated Akt, not S6K1 and 4EBP1, p-mTOR might come from mTORC2. The authors need to test whether CHDT increases mTORC2 activity in DEX-induced muscle atrophy by immunoprecipitating with an anti-rictor and testing mTOR phosphorylation in IPed mTOR. 

Response 3: In Figure 4, CHDT activated not only Akt but also S6K1 and 4EBP1. Therefore, p-mTOR might come from mTORC1, and strictly speaking, we claim that CHDT increased the activity of mTORC1, which regulates protein synthesis and associates with increasing skeletal muscle mass. We also added the following to the ‘Introduction’ and ‘Discussion’ section:

‘Introduction’ section:

“The mTOR complex is biochemically distinguished into mTOR complex 1 (mTORC1) and mTOR complex 2 (mTORC2). Both complexes have mTOR as a common subunit, each with its own unique components. Among them, mTORC1 is known to be a key regulator of skeletal muscle mass. mTORC1 upregulates protein translation through the phosphorylation of eukaryotic translation initiation factor 4E-binding protein 1 (4EBP1) and ribosomal protein S6 kinase beta-1 (S6K1), leading to cell growth and proliferation.”

‘Discussion’ section:

“However, CHDT administration dose-dependently increased Akt phosphorylation and prevented the impairments in downstream signaling of mTOR. Activation of S6K1 and 4EBP1 by CHDT administration resulted from regulation of mTORC1, implying that CHDT contributed to upregulating protein translation and muscle cell proliferation in skeletal muscle.”

Point 4: The authors need to test the effect of rapamycin on CHDT-induced prevention of DEX-induced muscle.  

Response 4: Thanks for your comments. Unfortunately, however, the efficacy of rapamycin on CHDT-induced prevention of Dex-induced muscle cannot be tested due to financial problems and all experimental animals have been sacrificed. Please understand our situation. We’ll reflect your opinion if further research on the muscle strength improvement effect of CHDT is conducted.

Point 5: When did you measure the lean mass and muscle mass of mice with DEX-induced muscle atrophy in Fig. 3? You need to add detailed information (duration for treatment; the number of mice for a group) in all Figure legends.

Response 5: According to your comments, we added detailed information to the legend of Figure 3. The administration period and details of the experimental protocol were added to the legend of Figure 2A.

Revised legend of Figure 3:

“Figure 3. Effects of CHDT on the lean mass and muscle mass of mice with Dex-induced muscle atrophy. (A) Body composition images obtained using DXA analysis. Lean tissue is shown in green, fat in red, and bone in white. The lean mass of the mice was measured twice: before the start of the study and at the end of the period of administration (on day 30). (B) Lean mass and (C) lean per-centage in the ROI, which surrounds the mouse hindlimbs. (D) At the end of the study, representative images of the quadriceps (QUA, top) and gastrocnemius (GAS, bottom) muscles were taken. The scale of one grid is 1 mm. (E) Masses of the QUA (left) and GAS (right) muscles, normalized to the most recently measured body mass. Different letters indicate statistically significant differences; p < 0.05, a > b > c.”

Revised legend of Figure 2:

“Figure 2. Effects of CHDT on the body mass and muscle strength of mice with Dex-induced muscle atrophy. (A) Experimental protocol. CHDT was orally administered daily to 6-week-old mice for 30 days, and from 14 days after the start of administration, Dex was IP injected daily. (B) Body mass and (C) grip strength of the mice were measured on days 0, 7, 14, 18, 22, 26, and 30.“ns” indicates that there was no statistically significant difference between all groups. (p>0.05). Different letters indi-cate statistically significant differences; p < 0.05, a > b > c > d.”

Point 6: The effect of CHDT on MyoD and myogenin in Qua was different from one in Gas. The authors need to discuss this in ‘Discussion’ section. 

Response 6: According to your comment, we added the following to the ‘Discussion’ section:

“This effect may also be mediated through the prevention of the reduction in activity of the myogenic transcription factors MyoD and myogenin, causing a restoration of muscle fiber size. We found that Dex treatment suppressed the expression of MyoD and myogenin, but was significantly increased by CHDT treatment. These results were seen in both muscles, especially in GAS rather than QUA.”

Point 7: The authors need to move the result of HPLC from the materials and methods section to the result section. 

Response 7: According to your comment, we moved the result of HPLC from the materials and methods section to the result section (Result 3.1).

Point 8: All data points need to be presented in all figures.

Response 8: Data points were added.

Reviewer 2 Report

The manuscript presents the positive effect of herbs extracts from A gigas and A dracunculus on Dex-induced muscle atrophy. This paper is well-written, of interest and shows a strong amelioration and prevention of muscle atrophy also highlighting many molecular players involved. The investigation is clear and linear even though a wider screening and omic approaches would have strenghtned the significance.

I have just few points to deepen:

-In figure 1 the 7-methoxycoumarin peak is not the main peak in the panel as indicated in the text; what about the high peak at 20 minutes?

-In 2.6 section the details of the antibodies used in western blot are lacking.

-How the authors chose the two dosages used in the experiments?

-Have the authors tested the effect of CHDT supplementation in control mice over time? This would be of interest to understand the potential benefits on normal subjects.

-Considering the pivotal role of mTOR in the autophagy and mitophagy activation and the early stimulation of autophagy in Dex treated myotubes, did the authors investigate the autophagy involvement in the CHDT induced effects (doi: 10.4161/cc.29272)? This besides the effect of mTOR phosphorylation on 4EBP1. Moreover, the mitochondrial involvement in the above processes may represent an important hub since mitochondrial dynamics could be associated with the atrophy program. Accordingly, the use of mitochondrial fission inhibitors such as mdivi1, can influence atrophy genes (doi: 10.4161/cc.29272) besides influencing and reshaping the mitochondrial network (DOI: 10.1038/s41418-020-0510-7) and thus potentially mitophagy (DOI: 10.1002/jcsm.12794; DOI: 10.1371/journal.pone.0032388). Please discuss with proper references.

Author Response

We sincerely thank the Reviewer for their thorough review of our manuscript and for the constructive suggestions that we received. We have tried to adequately respond to each comment received from the Reviewer. We believe that the Reviewer’s comments and suggestions have considerably improved our manuscript. We hope that our manuscript will be published in the Cells.

Comments from Reviewer 2

Point 1: In figure 1 the 7-methoxycoumarin peak is not the main peak in the panel as indicated in the text; what about the high peak at 20 minutes?

Response 1: Thanks for your comment. The analysis method was established by referring to several papers that revealed the efficacy of 7-methoxycoumarin isolated from Artemisia dracunculus [1, 2], and the peak of 7-methoxycoumarin in the HPLC chromatogram was confirmed and measured the content. In Korea, in accordance with the regulations on the recognition of functional materials for health functional foods, it is important to determine the “index compound” among the compounds contained in raw materials for the purpose of quality control. Accordingly, we set 7-methoxycoumarin as an index compound by referring to several papers showed various bioactive effects of 7-methoxycoumarin [1-3], and only performed HPLC analysis on it. It is not known exactly about the main peak; we speculate that the main peak is a type of tannin or flavonoid known to be abundantly contained in Artemisia dracunculus. Words that may cause confusion were deleted and sentences were edited.

In 3.1 section: “A representative HPLC chromatogram for the CHDT is shown as Figure 1. The peaks of decursin (the index component of A. gigas) and 7-methoxycoumarin (the index component of A. dracunculus) were identified, and their concentrations were analyzed.”

  1. Surveswaran, S.; Cai, Y. Z.; Xing, J.; Corke, H.; Sun, M., Antioxidant properties and principal phenolic phytochemicals of Indian medicinal plants from Asclepiadoideae and Periplocoideae. Nat Prod Res 2010, 24, (3), 206-21.
  2. Aydin, T.; Akincioglu, H.; Gumustas, M.; Gulcin, I.; Kazaz, C.; Cakir, A., human monoamine oxidase (hMAO) A and hMAO B inhibitors from Artemisia dracunculus L. herniarin and skimmin: human mononamine oxidase A and B inhibitors from A. dracunculus L. Z Naturforsch C J Biosci 2020, 75, (11-12), 459-466.
  3. Ekiert, H.; Swiatkowska, J.; Knut, E.; Klin, P.; Rzepiela, A.; Tomczyk, M.; Szopa, A., Artemisia dracunculus (Tarragon): A Review of Its Traditional Uses, Phytochemistry and Pharmacology. Front Pharmacol 2021, 12, 653993.

Point 2: In 2.6 section the details of the antibodies used in western blot are lacking.

Response 2: According to your comment, we added the details of the antibodies used in western blot in 2.6 section.

In 2.6 section: “The membranes were incubated overnight at 4°C with the primary antibodies; PI3K, p-Akt, Akt, p-mTOR, mTOR, p-p70S6K, p70S6K, p-4EBP1, 4EBP1, myoD, myogenin, p-FoxO3a, FoxO3a, Fbx32, and MuRF1, and then with a secondary antibody (peroxi-dase-conjugated anti-rabbit, anti-mouse, or anti-goat antibodies; Bio-Rad, Hercules, CA, USA) for 1 h at room temperature.”

Point 3: How the authors chose the two dosages used in the experiments?

Response 3:

The daily dosage of CHDT in mice was established by considering human intake. The dose administered to the mice was set to two concentrations, 350 and 500 mg/kg/day, and this corresponds to an adult daily intake of 1.75 g and 2.5 g per day which are appropriate doses for adult to take once a day as a natural supplement for functional food. A human clinical trial on the muscle strength improvement efficacy of CHDT in a dose within above range is in progress, and we aimed to elucidate the molecular mechanism of the muscle function improvement effect of CHDT through animal experiments. Also, some papers that published several bioactive effects of Angelica gigas and Artemisia dracunculus extract were helped to confirm the dosage determination [4-7].

  1. Song, Y. R.; Jang, B.; Lee, S. M.; Bae, S. J.; Bak, S. B.; Kim, Y. W., Angelica gigas NAKAI and Its Active Compound, Decursin, Inhibit Cellular Injury as an Antioxidant by the Regulation of AMP-Activated Protein Kinase and YAP Signaling. Molecules 2022, 27, (6).
  2. Wang, J.; Fernandez, A. E.; Tiano, S.; Huang, J.; Floyd, E.; Poulev, A.; Ribnicky, D.; Pasinetti, G. M., An Extract of Artemisia dracunculus L. Promotes Psychological Resilience in a Mouse Model of Depression. Oxid Med Cell Longev 2018, 2018, 7418681.
  3. Eidi, A.; Oryan, S.; Zaringhalam, J.; Rad, M., Antinociceptive and anti-inflammatory effects of the aerial parts of Artemisia dracunculus in mice. Pharm Biol 2016, 54, (3), 549-54.
  4. Lee, H. J.; Lee, H. J.; Lee, E. O.; Lee, J. H.; Lee, K. S.; Kim, K. H.; Kim, S. H.; Lu, J., In vivo anti-cancer activity of Korean Angelica gigas and its major pyranocoumarin decursin. Am J Chin Med 2009, 37, (1), 127-42.

Point 4: Have the authors tested the effect of CHDT supplementation in control mice over time? This would be of interest to understand the potential benefits on normal subjects.

Response 4: Thanks for your opinion. We did not test the effect of CHDT supplementation in control mice because we investigated the “protective and preventive effects of CHDT” on muscle atrophy accompanying malnutrition, injury, aging and disease. This study demonstrated that the pre-administration of CHDT before the onset of the muscle atrophy effectively prevented the development of severe muscle atrophy. Considering that CHDT prevented muscle atrophy through activation of the protein synthesis pathway, continuous administration of CHDT to normal mice is expected to have a positive effect on muscle strength improvement.

Point 5: Considering the pivotal role of mTOR in the autophagy and mitophagy activation and the early stimulation of autophagy in Dex treated myotubes, did the authors investigate the autophagy involvement in the CHDT induced effects (doi: 10.4161/cc.29272)? This besides the effect of mTOR phosphorylation on 4EBP1. Moreover, the mitochondrial involvement in the above processes may represent an important hub since mitochondrial dynamics could be associated with the atrophy program. Accordingly, the use of mitochondrial fission inhibitors such as mdivi1, can influence atrophy genes (doi: 10.4161/cc.29272) besides influencing and reshaping the mitochondrial network (DOI: 10.1038/s41418-020-0510-7) and thus potentially mitophagy (DOI: 10.1002/jcsm.12794; DOI: 10.1371/journal.pone.0032388). Please discuss with proper references.

Response 5: Thank you for your constructive comments. It is clear that mTOR plays an important role in the regulation of autophagy, and in particular, mTORC1 is known to regulate mitochondrial biogenesis. Given that CHDT administration promoted the activation of mTORC1, we agree with your opinion that the involvement of mitochondria may be an important hub in the process of CHDT’s muscle atrophy prevention. Based on this, we tried to conduct further research using the mitochondrial fission inhibitors you suggested, but we were unable to do due to financial and situational difficulties. Instead, we focused on the FoxO3a signaling pathway, which is known to promote the autophagy signaling chain by inducing autophagy-associated proteins. As shown in Figure 6, FoxO3a activity induced by Dex treatment increased the expression of Fbx32 and MuRF1, which are highly correlated with autophagy process, but CHDT inhibited them. Therefore, we argue that CHDT effectively modulated the Dex-induced excessive autophagy.

Added the above to the ‘Discussion’ section with appropriate references as follows:

“FoxO3a-induced protein degradation is mediated through the ubiquitin-proteasome and autophagy-lysosomal system. [39, 40]. Adequate autophagy is known to be a re-quired process for maintaining muscle mass [41], but hyperactivity of catabolic pro-cess, including the ubiquitin-proteasome system and autophagy-lysosomal system, leads to muscle wasting [42]. High expression of the E3 ubiquitin ligases Fbx32 and MuRF1 is a major mediator of muscle wasting and is highly correlated with the pro-motion of autophagy in muscle cells [13, 43, 44]. As expected, expression of these two proteins was increased by Dex treatment, but CHDT increased the phosphorylation of FoxO3a, which would block its nuclear translocation, and reduced the expression of E3 ubiquitin ligases, thereby downregulating both systems involved in muscle degradation.”
